# RESEARCH

# The readability of scientific texts is decreasing over time

**Abstract** Clarity and accuracy of reporting are fundamental to the scientific process. Readability formulas can estimate how difficult a text is to read. Here, in a corpus consisting of 709,577 abstracts published between 1881 and 2015 from 123 scientific journals, we show that the readability of science is steadily decreasing. Our analyses show that this trend is indicative of a growing use of general scientific jargon. These results are concerning for scientists and for the wider public, as they impact both the reproducibility and accessibility of research findings.

DOI: https://doi.org/10.7554/eLife.27725.001

**PONTUS PLAVÉN-SIGRAY[†], GRANVILLE JAMES MATHESON[†], BJÖRN CHRISTIAN SCHIFFLER[†] AND WILLIAM HEDLEY THOMPSON\***

**\*For correspondence:** hedley@startmail.com

[†]These authors contributed equally to this work

**Competing interests:** The authors declare that no competing interests exist.

## Introduction

Reporting science clearly and accurately is a fundamental part of the scientific process, facilitating both the dissemination of knowledge and the reproducibility of results. The clarity of written language can be quantified using readability formulas, which estimate how understandable written texts are (*Flesch, 1948*; *Kincaid et al., 1975*; *Chall and Dale, 1995*; *Danielson, 1987*; *DuBay, 2004*; *Štajner et al., 2012*). Texts written at different times can vary in their readability: trends towards simpler language have been observed in US presidential speeches (*Lim, 2008*), novels (*Danielson et al., 1992*; *Jatowt and Tanaka, 2012*) and news articles (*Stevenson, 1964*). There are studies that have investigated linguistic trends within the scientific literature. One study showed an increase in positive sentiment (*Vinkers et al., 2015*), finding that positive words such as 'novel' have increased dramatically in scientific texts since the 1970s. A tentative increase in complexity has been reported in scientific texts in a limited dataset (*Hayes, 1992*), but the extent of this phenomenon and any underlying reasons for such a trend remain unknown.

To investigate trends in scientific readability over time, we downloaded 709,577 article abstracts from PubMed, from 123 highly cited journals selected from 12 fields of research (*Figure 1A–C*). These journals cover general,

biomedical and life sciences. This journal list included, among others, Nature, Science, NEJM, The Lancet, PNAS and JAMA (see Materials and methods and *Supplementary file 1*) and the publication dates ranged from 1881 to 2015. We quantified the reading level of each abstract using two established measures of readability: the Flesch Reading Ease (FRE; *Flesch, 1948*; *Kincaid et al., 1975*) and the New Dale-Chall Readability Formula (NDC; *Chall and Dale, 1995*). The FRE is calculated using the number of syllables per word and the number of words in each sentence. The NDC is calculated using the number of words in each sentence and the percentage of 'difficult words'. Difficult words are defined as those words which do not belong to a predefined list of common words (see Materials and methods). Lower readability is indicated by a low FRE score or a high NDC score (*Figure 1A*).

## Results

The primary research question was to examine how the readability of an article's abstract relates to its year of publication. We observed a strong decreasing trend of the average yearly FRE ($r = -0.93$, 95% CI [-0.95,-0.90], $p < 10^{-15}$) and a strong increasing trend of average yearly NDC ($r = 0.93$, 95% CI [0.91,0.95], $p < 10^{-15}$) (*Figure 2A–E*). Next, we examined the

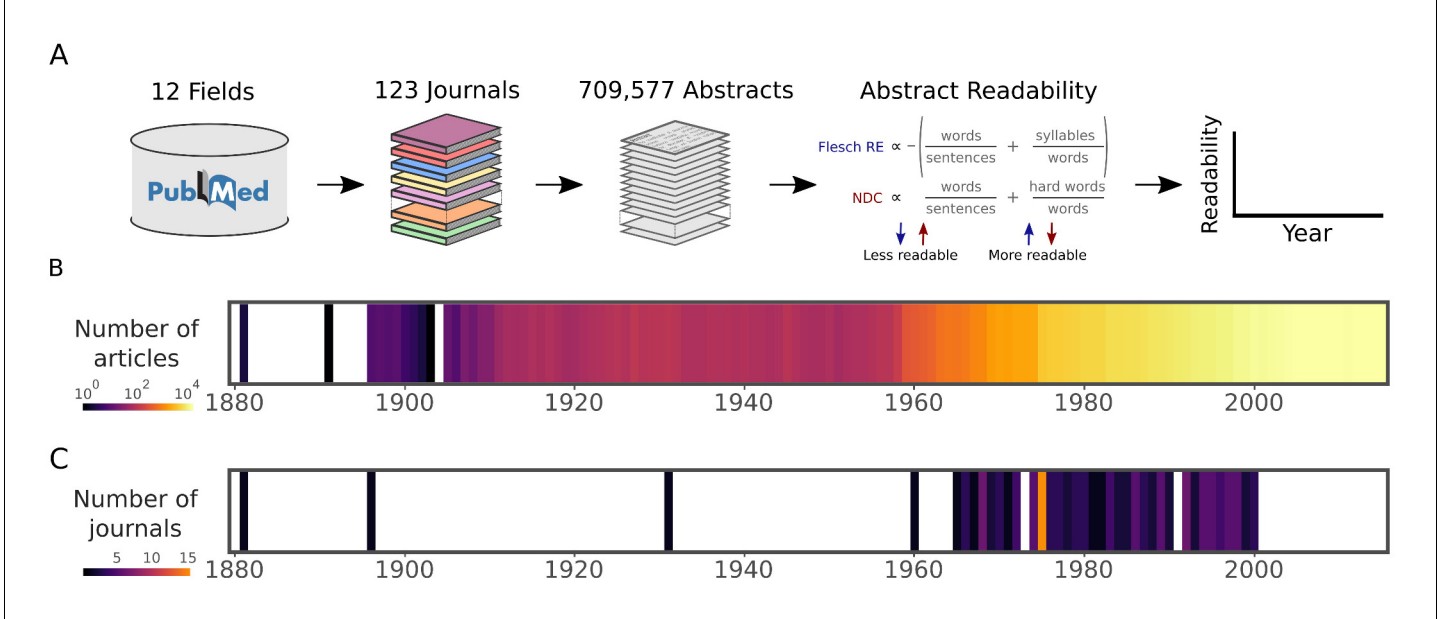

**Figure 1.** Data and readability analysis pipeline. (**A**) Schematic depicting the major steps in the abstract extraction and analysis pipeline. Readability formulas are provided in full in Materials and methods. (**B**) Number of articles in the corpus published in each year. The color scale is logarithmic. (**C**) Starting year of each journal within the corpus. This corresponds to the first article in PubMed with an abstract. The color scale is linear. Source data for this figure is available in *Figure 2—source data 1*.

DOI: https://doi.org/10.7554/eLife.27725.002

relationship between the components of the readability metrics and year of publication. The average number of syllables in each word (FRE component) and the percentage of difficult words (NDC component) showed pronounced increases over years (*Figure 2F,G*). Sentence length (FRE and NDC component) showed a steady increase with year after 1960 (*Figure 2H*), the period in which the majority of abstracts were published (*Figure 1B*). FRE and NDC were correlated with one another ($r = -0.72$, 95% CI [-0.72,-0.72], $p < 10^{-15}$) (*Figure 2E*).

The readability of individual abstracts was formally evaluated in relation to year of publication using a linear mixed effects model with journal as a random effect for both measures. The fixed effect of year was significantly related to FRE and NDC scores (*Table 1*). The average yearly trends combined with this statistical model reveal that the complexity of scientific writing is increasing with time. In order to explore whether this trend was consistent across all 12 selected fields, we extracted the slopes (random effects) from the mixed effects models for each journal. This showed that the trend of decreasing readability over time is not specific to any particular field, although there are differences in magnitude between fields (*Figure 3*). Further, only two journals out of 123 showed clear increases in

FRE across time (*Figure 3—figure supplement 1*).

To verify that the readability of abstracts was representative of the readability of the entire articles, we downloaded full text articles from six additional independent journals from which all articles were available in the PubMed Central Open Access Subset (*Figure 4A*). Although, as has previously been reported (*Dronberger and Kowitz, 1975*), abstracts are less readable than the full articles, there was a strong positive relationship between readability of the abstracts and the full texts (FRE: $r = 0.60$, 95% CI [0.60, 0.60], $p < 10^{-15}$; NDC: $r = 0.63$, 95% CI [0.63, 0.63], $p < 10^{-15}$, *Figure 4B*, *Figure 4—figure supplements 1* and *2*). This implies that the increasing complexity of scientific writing generalizes to the full texts.

There could be a number of explanations for the observed trend in scientific readability. We formulated two plausible and testable hypotheses: (1) There is an increase in the number of co-authors over time (*Figure 5A*) (see also (*Epstein, 1993*; *Drenth, 1998*). If the number of co-authors correlates with readability, this underlies the observed effect (i.e. a case of 'too many cooks spoil the broth'). (2) An increase in a general scientific jargon is leading to a

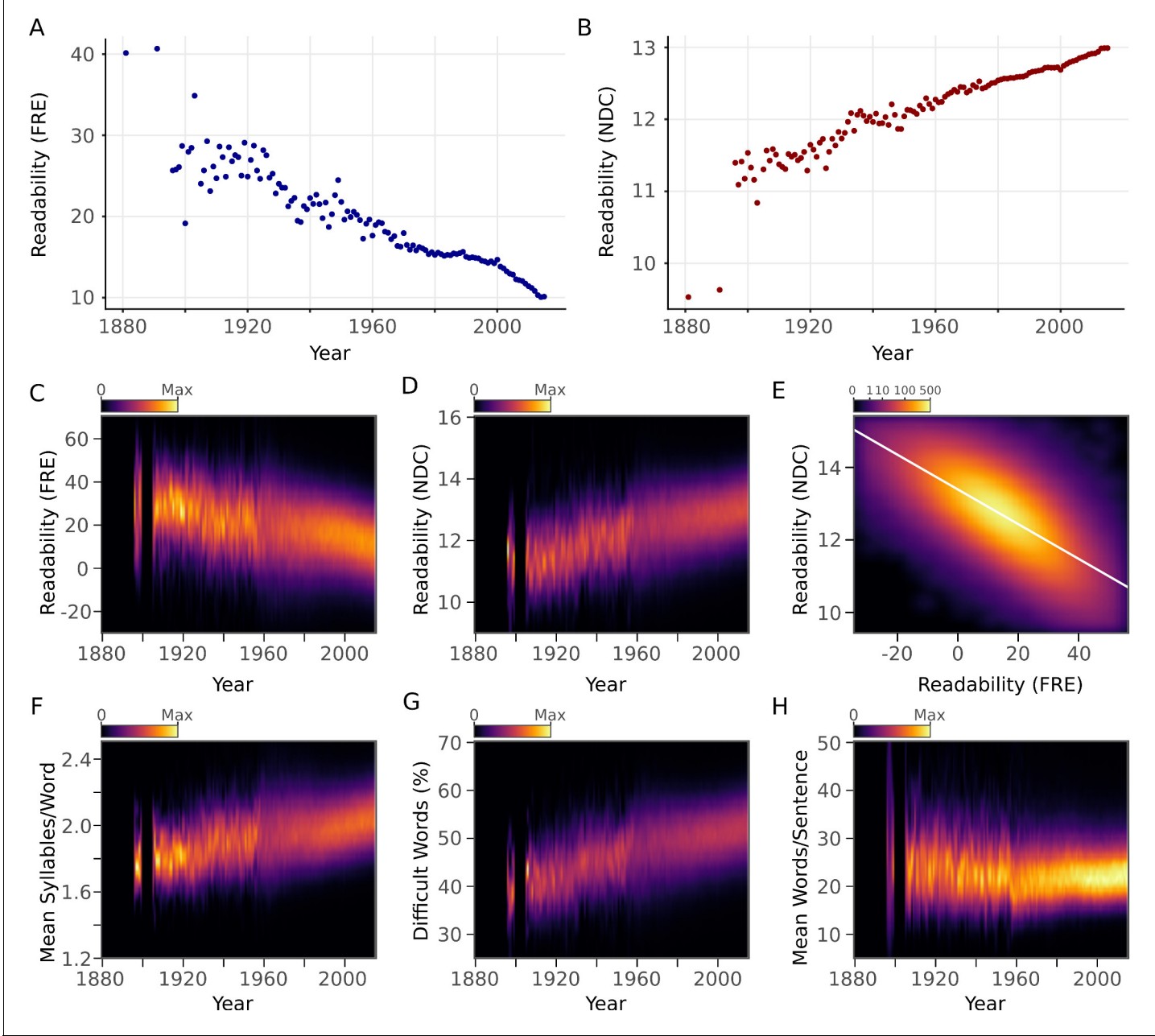

**Figure 2.** Scientific abstracts have become harder to read over time. (**A**) Mean Flesch Reading Ease (FRE) readability for each year. Lower scores indicate less readability. (**B**) Mean New Dale-Chall (NDC) readability for each year. Higher scores indicate less readability. (**C,D**) Kernel density estimates displaying the readability (C: FRE, D: NDC) distribution of all abstracts for each year. Color scales are linear and represent relative density of scores within each year. (**E**) Relationship between FRE and NDC scores across all abstracts, depicted by a two-dimensional kernel density estimate. Axis limits are set to include at least 99% of the data. The color scale is exponential and represents the number of articles at each pixel. (**F-H**) Kernel density estimates displaying the components of the readability measures (F: syllable to word ratio; G: percentage of difficult words; H: word to sentence ratio) distribution of all abstracts for each year. Color scales are linear and represent relative density of values within each year. For kernel density plots over time (**C,D,F,G,H**), years with fewer than 10 abstracts are excluded to obtain accurate density estimates.

DOI: https://doi.org/10.7554/eLife.27725.003

The following source data and figure supplement are available for figure 2:

**Source data 1.** Readability data of abstracts and number of authors per article.
DOI: https://doi.org/10.7554/eLife.27725.005

**Source data 2.** Readability data when no preprocessing is done.
DOI: https://doi.org/10.7554/eLife.27725.006

*Figure 2 continued*

**Figure supplement 1.** Readability over years with minimal preprocessing to illustrate that the preprocessing steps have not induced the trend.

DOI: https://doi.org/10.7554/eLife.27725.004

vocabulary which is almost exclusively used by scientists and less readable in general (i.e. a 'science-ese').

To test the first hypothesis, we divided the data by the number of authors. More authors were associated with decreased readability (*Figure 5B*, *Figure 5—figure supplement 1A*). However, we observed the same trend of decreasing readability across years regardless of the number of authors (*Figure 5C*, *Figure 5—figure supplement 1B*). When we included the number of authors as a predictor in the linear mixed effects model, it was significantly related to readability, while the fixed effect of year remained significant (*Table 2*). We can therefore reject the hypothesis that the increase in the number of authors on scientific articles is responsible for the observed trend, although abstract readability does decrease with more authors.

To test the second hypothesis, we constructed a measure for in-group scientific vocabulary. We selected the 2,949 most common words which were not included in the NDC common word list from 12,000 abstracts sampled at random (see Materials and methods for procedure). This is analogous to a 'science-specific common word list'. This list also includes topics which have increased over time (e.g. 'gene') and subject-specific words (e.g. 'tumor'), which are not indicative of an in-group scientific vocabulary. We removed such words to create a general scientific jargon list (2,138 words; see Materials and methods and *Supplementary file 2*). While the percentage of common words

from the NDC common word list decreased with year ($r = -0.93$, 95% CI [-0.93, -0.93], $p < 10^{-15}$, *Figure 6A*), there was an increase in the percentage of science-specific common words ($r = 0.90$, 95% CI [0.90, 0.91], $p < 10^{-15}$) and general scientific jargon ($r = 0.96$, 95% CI [0.95, 0.96], $p < 10^{-15}$) (*Figure 6A*). Twelve general science jargon words are presented in *Figure 6B*. While one word ('appears') decreased with time, all the remaining examples show sharp increases over time. Taken together, this provides evidence in favor of the hypothesis that there is an increase in general scientific jargon which partially accounts for the decreasing readability.

## Discussion

From analyzing over 700,000 abstracts in 123 journals from the biomedical and life sciences, as well as general science journals, we have shown a steady decrease of readability over time in the scientific literature. It is important to put the magnitude of these results in context. A FRE score of 100 is designed to reflect the reading level of a 10- to 11-year old. A score between 0 and 30 is considered understandable by college graduates (*Flesch, 1948*; *Kincaid et al., 1975*). In 1960, 14% of the texts in our corpus had a FRE below 0. In 2015, this number had risen to 22%. In other words, more than a fifth of scientific abstracts now have a readability considered beyond college graduate level English. However, the absolute readability scores should be interpreted with some caution: scores can vary due

**Table 1.** Model fits for two different linear mixed effect models examining the relationship between readability scores and year.

| Metric | Model | dAIC | dBIC | beta | CI 95% | t | df | p |
|---|---|---|---|---|---|---|---|---|
| FRE | M0 | 16008 | 15974 | - | - | - | - | - |
|  | M1 | 5240 | 5217 | -0.14 | [-0.15, -0.14] | -104.2 | 709543 | $p < 10^{-15}$ |
|  | M2 | 0 | 0 | -0.19 | [-0.22, -0.16] | -12.7 | 123 | $p < 10^{-15}$ |
| NDC | M0 | 28593 | 28559 | - | - | - | - | - |
|  | M1 | 4077 | 4054 | 0.014 | [0.014, 0.014] | 158.0 | 709559 | $p < 10^{-15}$ |
|  | M2 | 0 | 0 | 0.016 | [0.015, 0.018] | 20.5 | 117 | $p < 10^{-15}$ |

A null model (M0) without year as a predictor is included as a baseline comparison. Lower dAIC and dBIC values indicate better model fit. FRE = Flesch Reading Ease; NDC = New Dale-Chall Readability Formula; M0 = Journal as random effect with varying intercepts; M1 = M0 with an added fixed effect of time; M2 = M1 with varying slopes for the random effect of journal; dAIC = difference in Akaike Information Criterion from the best fitting model (M2); dBIC = difference in Bayesian Information Criterion from the best fitting model (M2); df = Degrees of Freedom calculated using Satterthwaite approximation.

DOI: https://doi.org/10.7554/eLife.27725.007

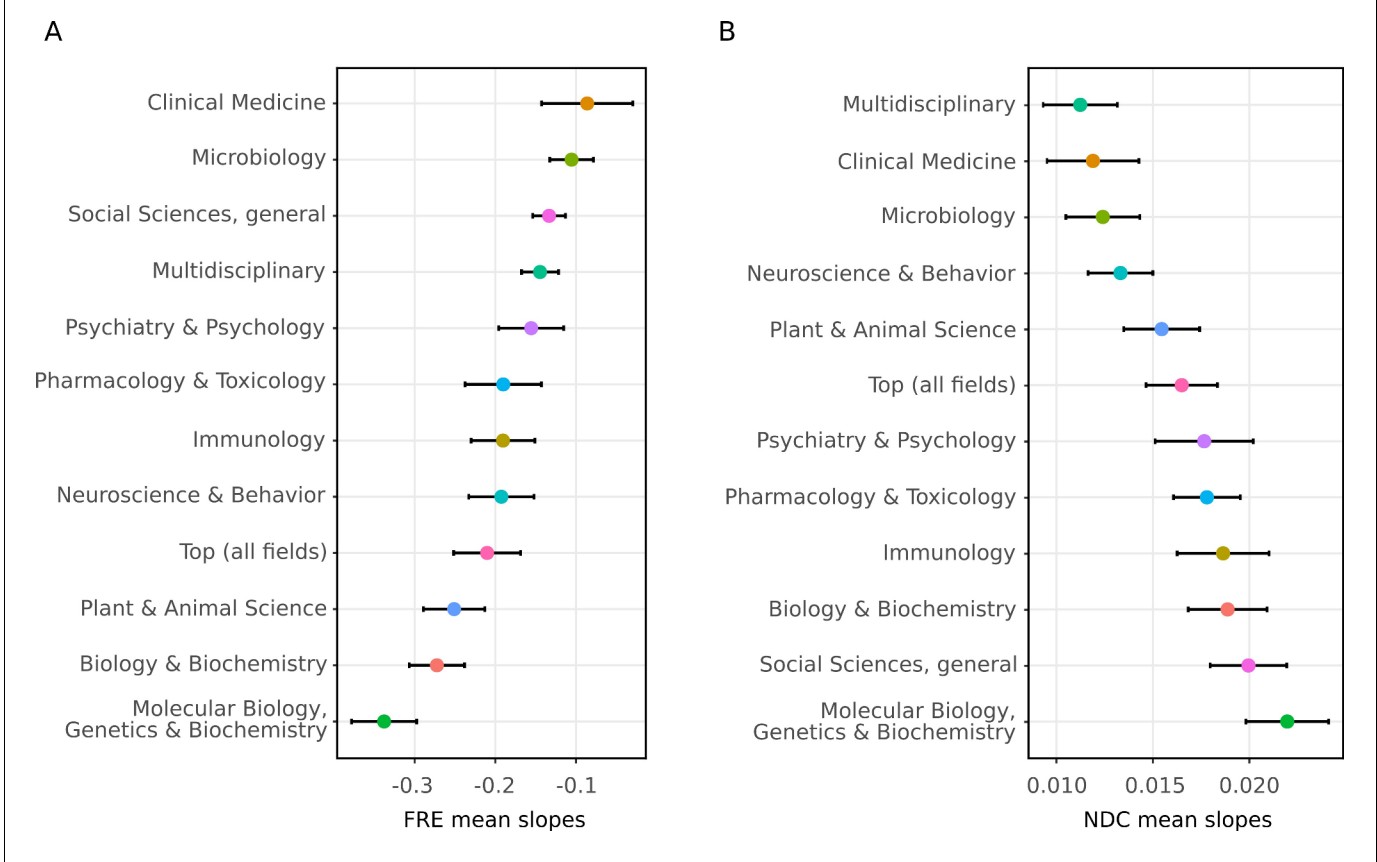

**Figure 3.** The decline in readability differs between scientific fields. The random slopes for each journal were extracted from the best fitting linear mixed effect model (M2) and summarized according to which field they belong to (The error bars represent SE of the mean slope). Since some journals belong to more than one field, some random slopes appear in more than one summary. The trend of decreasing readability is not specific to any one field. (**A**) Summaries of random slopes for Flesch Reading Ease. (**B**) Summaries of random slopes for New Dale-Chall.

DOI: https://doi.org/10.7554/eLife.27725.008

The following source data and figure supplement are available for figure 3:

**Source data 1.** Summary of FRE and NDC journal random slopes for each field extracted from the linear mixed model (M2).

DOI: https://doi.org/10.7554/eLife.27725.010

**Source data 2.** FRE and NDC random slopes for each journal extracted from the linear mixed model (M2).

DOI: https://doi.org/10.7554/eLife.27725.011

**Figure supplement 1.** Most, but not all, journals have become less readable over time.

DOI: https://doi.org/10.7554/eLife.27725.009

to different media (e.g. comics versus news articles; *Štajner et al., 2012*) and education level thresholds can be imprecise (*Stokes, 1978*). We then validated abstract readability against full text readability, demonstrating that it is a suitable approximation for comparing main texts.

We investigated two possible reasons why this trend has occurred. First, we found that readability of abstracts correlates with the number of co-authors, but this failed to fully account for the trend through time. Second, we showed that there is an increase in general scientific jargon over years. These general science jargon words should be interpreted as words which scientists frequently use in scientific texts, and not as subject specific jargon. This finding is indicative of a progressively increasing in-group scientific language ('science-ese').

An alternative explanation for the main finding is that the cumulative growth of scientific knowledge makes an increasingly complex language necessary. This cannot be directly tested, but if this were to fully explain the trend, we would expect a greater diversity of vocabulary as science grows more specialized. While accounting for the original finding of the

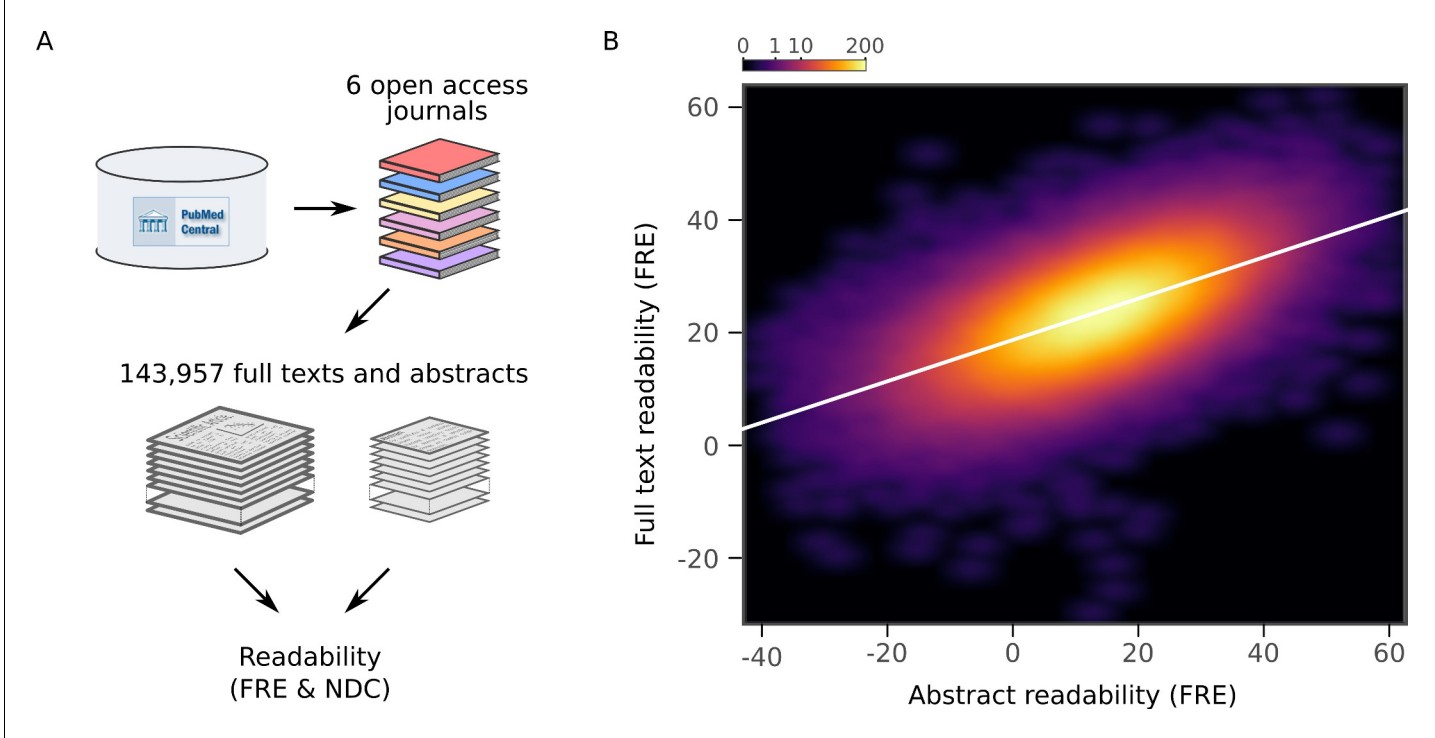

**Figure 4.** Readability of scientific abstracts correlates with readability of full texts. (**A**) Schematic depicting the major steps in the full text extraction and analysis pipeline. (**B**) Relationship between Flesch Reading Ease (FRE) scores of abstracts and full texts across the full text corpus, depicted by a two-dimensional kernel density estimate. The color scale is exponential and represents the number of articles at each pixel. Axis limits are set to include at least 99% of the data. For New Dale-Chall (NDC) scores, see *Figure 4—figure supplement 1*. For each journal separately, see *Figure 4—figure supplement 2*.

DOI: https://doi.org/10.7554/eLife.27725.012

The following source data and figure supplements are available for figure 4:

**Source data 1.** Readability data used in full text analysis.
DOI: https://doi.org/10.7554/eLife.27725.015
**Figure supplement 1.** New Dale-Chall abstracts and full text.
DOI: https://doi.org/10.7554/eLife.27725.013
**Figure supplement 2.** Correlations between readability metrics for abstracts and full texts from individual journals.
DOI: https://doi.org/10.7554/eLife.27725.014

increase in difficult words and of syllable count, this would *not* explain the increase of general scientific jargon words (e.g. 'furthermore' or 'novel', *Figure 6B*). Thus, this possible explanation cannot fully account for our findings.

Lower readability implies less accessibility, particularly for non-specialists, such as journalists, policy-makers and the wider public. Scientific journalism offers a key role in communicating science to the wider public (*Bubela et al., 2009*) and scientific credibility can sometimes suffer when reported by journalists (*Hinnant and Len-Ríos, 2009*). Considering this, decreasing readability cannot be a positive development for efforts to accurately communicate science to non-specialists. Further, amidst concerns that modern societies are

becoming less stringent with actual truths, replaced with true-sounding 'post-facts' (*Manjoo, 2011*; *Nordenstedt and Rosling, 2016*) science should be advancing our most accurate knowledge. One way to achieve this is for science to maximize its accessibility to non-specialists.

Lower readability is also a problem for specialists (*Hartley, 1994*; *Hartley and Benjamin, 1998*; *Hartley, 2003*). This was explicitly shown by *Hartley (1994)* who demonstrated that rewriting scientific abstracts, to improve their readability, increased academics' ability to comprehend them. While science is complex, and some jargon is unavoidable (*Knight, 2003*), this does not justify the continuing trend that we have shown. It is also worth considering the

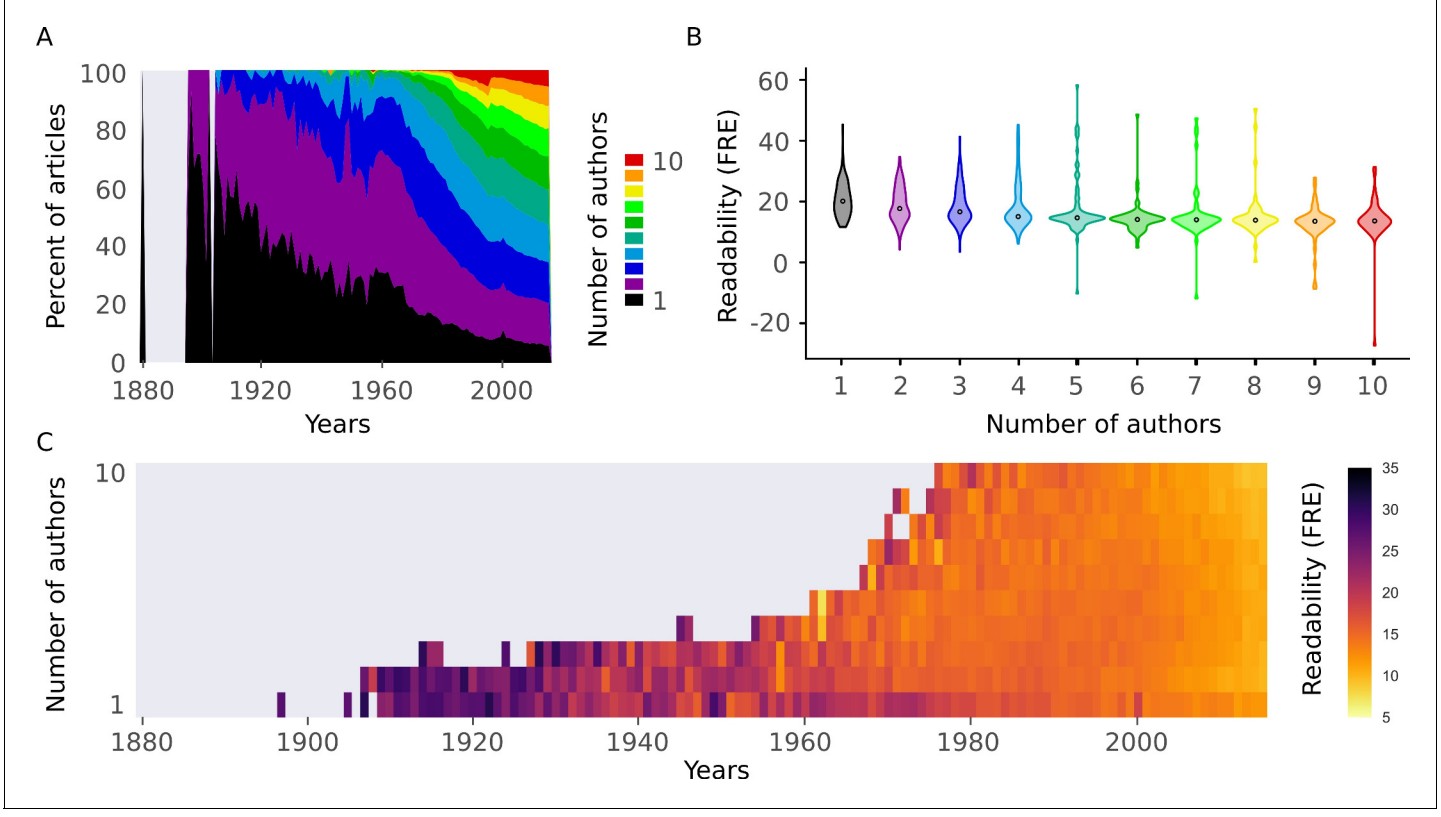

**Figure 5.** Readability is affected by the number of authors. (A) Proportion of number of authors per year for all articles in the abstract corpus. (B) Distributions of Flesch Reading Ease (FRE) scores for different numbers of authors (1-10). For New Dale-Chall (NDC), see *Figure 5—figure supplement 1A* (C) Mean FRE score for each year for different numbers of authors (1-10). For visualization purposes, bins with fewer than 10 abstracts are excluded. For NDC, see *Figure 5—figure supplement 1B*. Source data for this figure is available in *Figure 2—source data 1*.
DOI: https://doi.org/10.7554/eLife.27725.016

The following figure supplement is available for figure 5:

**Figure supplement 1.** New Dale-Chall for different number of authors.
DOI: https://doi.org/10.7554/eLife.27725.017

importance of comprehensibility of scientific texts in light of the recent controversy regarding the reproducibility of science (*Prinz et al., 2011*; *McNutt, 2014*; *Begley and Ioannidis, 2015*; *Nosek et al., 2015*; *Camerer et al., 2016*). Reproducibility requires that findings can be verified independently. To achieve this, reporting of methods and results must be sufficiently understandable.

Readability formulas are not without their limitations. They provide an estimate of a text's readability and should not be interpreted as a categorical measure of how well a text will be understood. For example, readability can be affected by text size, line spacing, the use of headers, as well as by the use of visual aids such as tables or graphs, none of which are captured by readability formulas (*Hartley, 2013*; *Badarudeen and Sabharwal, 2010*). Many semantic properties of texts are overlooked,

including the complexity of ideas, the rhetorical structure and the overall coherence of the text (*Bruce et al., 1981*; *Danielson, 1987*; *Zamanian and Heydari, 2012*). Changing a text solely to improve readability scores does not automatically make a text more understandable (*Duffy and Kabance, 1982*; *Redish, 2000*).

Despite the limitations of readability formulas, our study shows that recent scientific texts are, on average, less readable than older scientific texts. This trend was not specific to any one field, even though the size of this association varied across fields. Some fields also had a steeper decline in NDC scores, while less of a decline in FRE scores, and vice versa (*Figure 3*). Further research should explore possible reasons for these differences, as it may give clues on how to improve readability. For example, the adoption of structured abstracts which are known to assist readability (*Hartley and*

**Table 2.** Linear mixed effect models predicting readability scores by year and number of authors with journals as random effect.

| Metric | Subset | n | Random Effect | beta | CI 95% | t | df | p |
|---|---|---|---|---|---|---|---|---|
| FRE | Yes* | 652357 | Year | -0.17 | [-0.19, -0.14] | -11.3 | 122 | $p<10^{-15}$ |
| | | | Authors | -0.24 | [-0.26, -0.23] | -30.0 | 651832 | $p <10^{-15}$ |
| | No | 707250 | Year | -0.18 | [-0.21, -0.15] | -12.3 | 123 | $p <10^{-15}$ |
| | | | Authors | -0.07 | [-0.08, -0.06] | -23.5 | 704922 | $p<10^{-15}$ |
| NDC | Yes* | 652357 | Year | 0.014 | [0.012, 0.015] | 16.5 | 119 | $p<10^{-15}$ |
| | | | Authors | 0.033 | [0.032, 0.034] | 63.6 | 651516 | $p <10^{-15}$ |
| | No | 707250 | Year | 0.016 | [0.014, 0.017] | 19.6 | 118 | $p <10^{-15}$ |
| | | | Authors | 0.008 | [0.007, 0.008] | 40.3 | 701014 | $p <10^{-15}$ |

FRE = Flesch Reading Ease; NDC = New Dale-Chall Readability Formula; df = Degrees of Freedom calculated using Satterthwaite approximation. ∗ signifies that abstracts with only 1 to 10 authors are included in the model.

DOI: https://doi.org/10.7554/eLife.27725.018

Benjamin, 1998; Hartley, 2003) might lead to a less steep decline for some fields.

What more can be done to reverse this trend? The emerging field of science communication deals with ways science can effectively communicate ideas to a wider audience (Treise and Weigold, 2002; Nielsen, 2013; Fischhoff, 2013). One suggestion from this field is to create accessible 'lay summaries', which have been implemented by some journals (Kuehne and Olden, 2015). Others have noted that scientists are increasing their direct communication with the general public through social media (Peters et al., 2014), and this trend could be encouraged. However, while these two suggestions may increase accessibility of scientific results, neither will reverse the readability trend of scientific texts. Another proposal is to make scientific communication a necessary part of undergraduate and graduate education (Brownell et al., 2013).

Scientists themselves can estimate their own readability in most word processing software. Further, while some journals aim for high readability, perhaps a more thorough review of article readability should be carried out by journals in the review process. Finally, in an era of data metrics, it is possible to assess a scientist's average readability, analogous to the h-index for citations (Hirsch, 2005). Such an 'r-index' could be considered an asset for those scientists who emphasize clarity in their writing.

## Materials and methods

### Journal selection
We aimed to obtain journals from which articles are highly cited from a representative selection of the biomedical and life sciences, as well as from journals which cover all fields of science, which were indexed on PubMed. Using the Thomson Reuters Research Front Maps (http://archive.sciencewatch.com/dr/rfm/) and the Thomson Reuters Journal Citation Reports, we selected 12 fields:

1. Top (all fields)
2. Biology & Biochemistry
3. Clinical Medicine
4. Immunology
5. Microbiology
6. Molecular Biology, Genetics & Biochemistry
7. Multidisciplinary
8. Neuroscience & Behavior
9. Pharmacology & Toxicology
10. Plant & Animal Science
11. Psychiatry & Psychology
12. Social Sciences, general

'Multidisciplinary' accounts for journals which publish work from multiple fields, but which did not fit into any one category. 'Top (all fields)' refers to the journals which are most highly cited across all fields, i.e. which have the highest impact factor from the 2014 Thomson Reuters Journal Citation Reports. The 'Social Sciences, general' field within the biomedical and life sciences includes journals from the subfields of 'Health Care Sciences & Services' and 'Primary Health Care'. Journals were semi-automatically selected by querying the PubMed API using the R package RISmed (Kovalchik, 2016) according to the following criteria:

1. There should be more than 15 years between the years of the first five and most recent five PubMed entries.
2. There should not be fewer than 100 articles returned for the journal.
3. The impact factor of the journal should not be below 1 according to the 2014

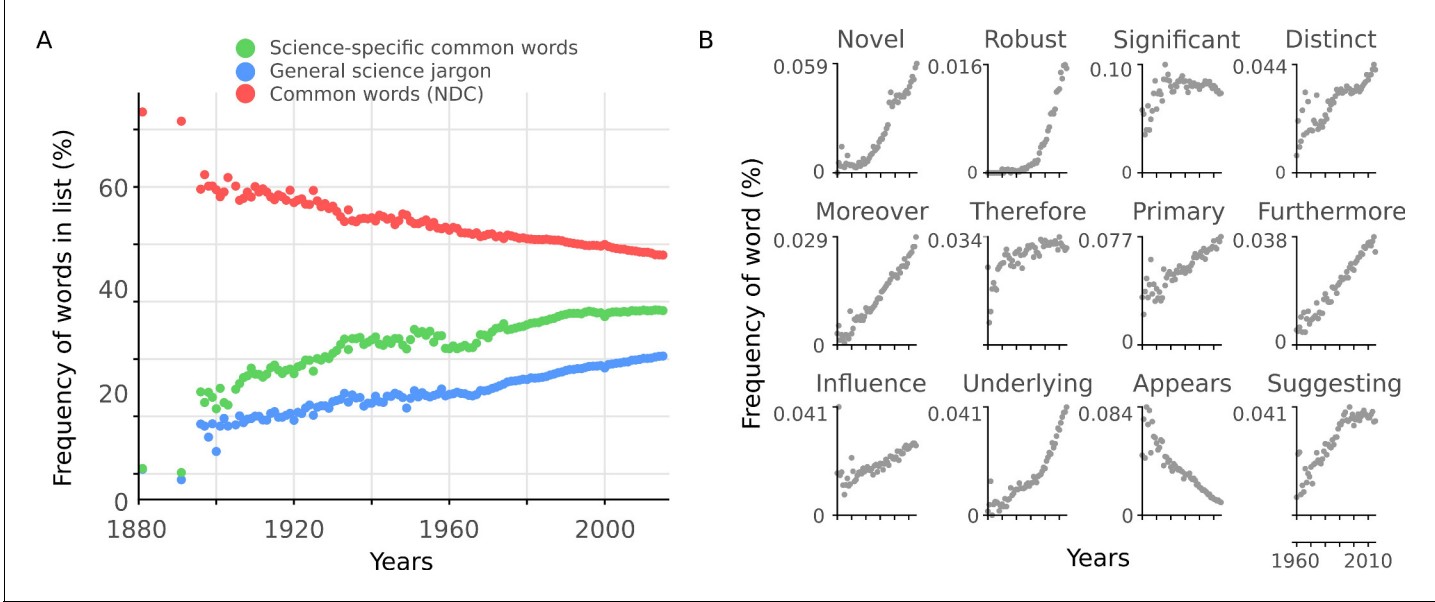

**Figure 6.** Readability is affected by general scientific jargon. (**A**) Mean percentage of words in abstracts per year included in three different lists: science-specific common words (green, 2,949 words), general scientific jargon (blue, 2,138 words) and NDC common words (red, 2,949 words). (**B**) Example general science jargon words taken from the general scientific jargon list. Mean percentage of each word's frequency in abstracts per year is shown.

DOI: https://doi.org/10.7554/eLife.27725.019
The following source data is available for figure 6:

**Source data 1.** Frequency of words in lists and example word use per article.
DOI: https://doi.org/10.7554/eLife.27725.020
**Source data 2.** PubMed ID for files used in training and verification lists of science common word list.
DOI: https://doi.org/10.7554/eLife.27725.021

Thomson Reuters Journal Citation Report.

4. The articles within the journal should be in English.
5. The number of selected journals should provide as equal representation as possible of subfields within the broader research fields.

From each of 11 of the fields, the 12 most highly cited journals were selected. The final field (Multidisciplinary) only contained six journals, as no more journals could be identified which met all inclusion criteria. Some journals exist in multiple fields, thus the number of journals (123) is below the possible maximum of 138 journals. See *Supplementary file 1* for the journals and their field mappings.

Articles were downloaded from PubMed between April 22, 2016 and May 15, 2016, and on June 12, 2017. The later download date was to correct for originally having only included 11 journals in one of the fields (when the data was first downloaded). The text of the abstract, journal name, title of article, PubMed IDs and publication year were extracted. Throughout the article, we only used data up to and including the year 2015.

*Language preprocessing*

Abstracts downloaded from PubMed were preprocessed so that the words and syllables could be counted. TreeTagger (version 3.2 (Linux); (*Schmid, 1994*) was used to identify sentence endings and to remove non-words (e.g. numbers) and any remaining punctuation from the abstracts. Scientific texts contain numerous phrasings which TreeTagger did not parse adequately. We did three rounds of quality control where at least 200 preprocessed articles, sampled at random, were compared with their original texts. After identifying irregularities with the TreeTagger performance, regular expression heuristics were created to prepare the abstracts prior to using the TreeTagger algorithm. After the three rounds of quality control, the stripped abstracts contained only words with at least one syllable and periods to end sentences. Sentences containing only one word were ignored.

The heuristic rules after quality control rounds included: removing all abbreviations, adding spaces after periods when missing, adding a final period at the end of the abstract when missing, removing numbers that ended sentences, identifying sentences that end with 'etc.' and keeping the period, removing all single letter words except 'a', 'A' and 'I', removing nucleic acid sequences, replacing hyphens with a space, removing periods arising from the use of binomial nomenclature, and removing copyright and funding information. All preprocessing scripts are available at https://github.com/wiheto/readabilityinscience (*Plavén-Sigray et al., 2017*; copy archived at https://github.com/elifesciences-publications/readabilityinscience). Examples of texts before and after preprocessing are presented in *Supplementary file 3*. We confirmed that the observed trends were not induced by the preprocessing steps by running the readability analysis presented in *Figure 1D,E* using the raw data (*Figure 2—figure supplement 1*).

### Language and readability metrics

Two well-established readability measures were used throughout the article: the Flesch Reading Ease (FRE) (*Flesch, 1948*; *Kincaid et al., 1975*) and the New Dale-Chall Readability Formula (NDC) (*Chall and Dale, 1995*). These measures use different language metrics: syllable count, sentence count, word count and percentage of difficult words. Two different readability metrics were chosen to ensure that the results were not induced by a single method. FRE was chosen due to its popularity and consistency with other readability metrics (*Didegah and Thelwall, 2013*), and because it has previously been applied to trends over time (*Lim, 2008*; *Danielson et al., 1992*; *Jatowt and Tanaka, 2012*; *Stevenson, 1964*). NDC was chosen since it is both well established and compares well with more recent methods for analyzing readability (*Benjamin, 2012*).

Counting the syllables of a word was performed in a three step fashion. First, the word was required to have a vowel or a 'y' in it. Second, the word was queried against a dictionary that contained specified syllable counts using the natural language toolkit (NLTK) (version 3.2.2; *Bird et al., 2009*). If there were multiple possible syllable counts for a given word, the longer alternative was chosen. Third, if the word was not in the dictionary, the number of vowels (excluding diphthongs) was counted. If a word

ended in a 'y', this was counted as an additional syllable in this third step.

Word count was calculated by counting all the words in the abstract that had at least one syllable. The number of sentences was calculated by counting the number of periods in the preprocessed abstracts.

The percentage of difficult words originated from (*Chall and Dale, 1995*), defined as words which do not belong to a list of common words. The 'NDC common word' list used here was taken from the NDC implementation in the textstat python package (https://github.com/shivam5992/textstat) which included 2,949 words (*Supplementary file 2*). This list excludes some words from the original NDC common word list (such as abbreviations, e.g. 'A.M.'; and double words, e.g. 'all right').

FRE uses both the average number of syllables per word and the average number of words per sentence to estimate the reading level.

$$FRE = 206.835 - 1.015\left(\frac{\text{words}}{\text{sentences}}\right) - 84.6\left(\frac{\text{syllables}}{\text{words}}\right)$$

where 'words', 'sentences' and 'syllables' entail the number of each in the text, respectively.

NDC scores are calculated by using the percentage of difficult words and the average sentence length of abstracts. While the NDC was originally calculated on 100 words due to computational limitations, we used the entire text.

$$NDC = \begin{cases} 0.1579\left(\frac{\text{difficult}}{\text{words}} \times 100\right) + 0.0496\left(\frac{\text{words}}{\text{sentences}}\right) + 3.6365 & \text{if} \left(\frac{\text{difficult}}{\text{words}}\right) > 5 \\ 0.1579\left(\frac{\text{difficult}}{\text{words}} \times 100\right) + 0.0496\left(\frac{\text{words}}{\text{sentences}}\right) & \text{if} \left(\frac{\text{difficult}}{\text{words}}\right) \leq 5 \end{cases}$$

where 'words', and 'sentences' entail the number of each in the text, respectively. 'Difficult' is the number of words that are not present in the NDC common word list.

We have used two well-established readability formulas in our analysis. The application of readability formulas has previously been questioned (*Duffy and Kabance, 1982*; *Redish and Selzer, 1985*; *Zamanian and Heydari, 2012*) and modern alternatives have been proposed (see (*Benjamin, 2012*). However, NDC has been shown to perform comparably with these more modern methods (*Benjamin, 2012*).

### Science-specific common word and general science jargon lists

We created two common word lists using the abstracts in our dataset:

*Science-specific common words*: Words frequently used by scientists which are not part of

the NDC common word list. This contains units of measurement (e.g. 'mol'), subject-specific words (e.g. 'electrophoresis'), general science jargon words (e.g. 'moreover') and some proper nouns (e.g. 'european').

*General science jargon*: A subset of science-specific common words. These are non-subject-specific words that are frequently used by scientists. This list contains words with a variety of different linguistic functions (e.g. 'endogenous', 'contribute' and 'moreover'). General science jargon can be considered the basic vocabulary of a '*science-ese*'. Science-ese is analogous to *legalese*, which is the general technical language used by legal professionals.

To construct the science-specific common word list, 12,000 articles were selected to identify words frequently used in the scientific literature. In order to avoid any recency bias, 2,000 articles were randomly selected from six different decades (starting at the 1960s). From these articles, the frequency of all words was calculated. After excluding words in the NDC common word list, the 2,949 most frequent words were selected. The number 2,949 was selected to be the same length as the NDC common word list.

To validate that this list is identifying a general scientific terminology, we created a verification list by performing the same steps as above on an additional independent set of 12,000 articles. Of the 2,949 words in the science-specific common word list, 90.95% of the words were present in the verification list (see *Supplementary file 2* for both word lists). The 24,000 articles used in the derivation and verification of the lists were excluded from all further analysis.

The general scientific jargon list contained 2,138 words. It was created by manually filtering the science-specific common word list. All four co-authors went through the science-specific common word list and rated each word. The following guidelines were formulated to exclude words from being classed as general science jargon: (1) abbreviations, roman numerals, or units that survived preprocessing (e.g. 'mol'). (2) Field-specific words (e.g. 'hepatitis'). (3) Words whose frequency may be changing through time due to major discoveries (e.g. 'gene'). (4) Nouns and adjectives that refer to non-science objects and could be placed in a general easy word list (e.g. 'mouse', 'green', 'September'). Remaining words were classed as possible general science jargon word. After comparing each co-authors' ratings, the final list was created. The co-authors

performed the ratings to identify jargon words independently. However, half way through the ratings there was a meeting to control that the guidelines were being performed in a similar way. In this meeting, the authors discussed examples from their ratings. Due to this, the ratings can not be classed as completely independent.

## Comparison of full texts vs abstracts

To compare the readability of full texts and abstracts, we chose six journals from the PubMed Central Open Access Subset for which all full texts of articles were available under a Creative Commons or similar license. The journals were BMC Biology, eLife, Genome Biology, PLoS Biology, PLoS Medicine and PLoS ONE. None of them were a part of the original journal list which was used in the main analysis. They were selected as they all cover biomedicine and life sciences and as open access journals, we were legally allowed to bulk-download both abstracts and full texts. However, none of the included journals have existed for a long period of time (*Supplementary file 4*). As such, they cannot be said to represent the same time range covered by the 123 journals used in the main analysis. Custom scripts were written to extract the full text in the textfiles downloaded from each respective journal. In total, 143,957 articles were included in the full text analysis. Both article abstracts and full texts were preprocessed according to the procedure outlined above and readability measures were calculated.

## Statistics

All statistical modeling was performed in R version 3.3.2.

We evaluated the relationship between the readability of single abstracts and year of publication separately for FRE and NDC scores. The data can be viewed as hierarchically structured since abstracts belonging to different journals may differ in key aspects. In addition, journals span over different ranges of years (*Figure 1C* and *Supplementary file 1*). In order to account for this structure, we performed linear mixed effect modeling using the R-packages lme4 version 1.1-12 (*Bates et al., 2014*), and lmerTest version 2.0-33 (*Kuznetsova et al., 2015*) with maximum likelihood estimation. We compared different models of increasing complexity. The included models were as following: (M0) a null model in which readability score was predicted

only by journal as random effect with varying intercepts; (M1) the same as M0, but with an added fixed effect of time; and (M2) the same as M1, but with varying slopes for the random effect of journal (*Table 1*). We selected the best fitting model as determined by the Akaike Information Criterion and the Bayesian Information Criterion.

In order to test that the trend was not explained by the increasing number of authors with year, we specified an additional model. It was identical to M2 above, but also included the number of authors as a second fixed effect. Some articles (n = 2,327) lacked author information, and were excluded from the analysis. This model was performed using two sets of the data: i) a subset including only articles with one to ten authors (n = 652,357), ii) a full dataset consisting of all articles with complete author information (n = 707,250) (*Table 2*). The motivation for (i) was that abstracts with many authors may bias the results.

### Acknowledgements

This article has a FRE score of 49. The abstract has a FRE score of 40.

**Pontus Plavén-Sigray** is in the Department of Clinical Neuroscience, Karolinska Institutet, Stockholm,Sweden

**Granville James Matheson** is in the Department of Clinical Neuroscience, Karolinska Institutet, Stockholm, Sweden

**Björn Christian Schiffler** is in the Department of Clinical Neuroscience, Karolinska Institutet, Stockholm, Sweden

ⓘ https://orcid.org/0000-0003-2347-4965

**William Hedley Thompson** is in the Department of Clinical Neuroscience, Karolinska Institutet, Stockholm, Sweden

hedley@startmail.com

ⓘ https://orcid.org/0000-0002-0533-6035

*Author contributions:* Pontus Plavén-Sigray, Granville James Matheson, Björn Christian Schiffler, Data curation, Formal analysis, Investigation, Visualization, Methodology, Writing—original draft, Writing—review and editing, Contributed to all aspects of the work. Contributed equally and their author order was determined randomly; William Hedley Thompson, Conceptualization, Data curation, Formal analysis, Investigation, Visualization, Methodology, Writing—original draft, Writing—review and editing, Conceived the study and wrote the majority of the analysis pipeline. Contributed to all aspects of the work

*Competing interests:* The authors declare that no competing interests exist.

### Additional files

#### Supplementary files

• Supplementary file 1. Journal information of abstract texts.
DOI: https://doi.org/10.7554/eLife.27725.022

• Supplementary file 2. Word lists used in analysis (NDC common words, Science common words, General science jargon, Verification list for science common words).
DOI: https://doi.org/10.7554/eLife.27725.023

• Supplementary file 3. Examples of language preprocessing on abstracts.
DOI: https://doi.org/10.7554/eLife.27725.024

• Supplementary file 4. Journal information for full text analysis.
DOI: https://doi.org/10.7554/eLife.27725.025

#### Major datasets

The following previously published dataset was used:

| Author(s) | Year | Dataset URL | Database, license, and accessibility information |
|---|---|---|---|
| PubMed | 2016 | ftp://ftp.ncbi.nlm.nih.gov/pub/pmc/oa_bulk/ | Publicly available at the NCBI ftp site for oa_bulk. |

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
