## [Decision Letter]

Thank you for submitting your article "The readability of scientific texts is decreasing over time" for consideration as a Feature Article by eLife. Your article has been reviewed by two peer reviewers, Ralf Barkemeyer (Reviewer #1) and Christiaan Vinkers (Reviewer #2), and an Associate Features Editor, Stuart King.

The reviewers have discussed the reviews with one another and the Associate Features Editor has drafted this decision to help you prepare a revised submission. We hope you will be able to submit the revised version within two months.

Summary

This manuscript – which first appeared as a preprint on bioRxiv – reports a decline in the readability of scientific texts meaning that, on average, papers published in recent decades are harder to read than those from over a century ago. Two possible explanations are tested: an increase in the number of authors per manuscript, and the wider use of uncommon language (referred to as "general scientific jargon"). All three reviewers agree that the manuscript is timely and relevant, and that it has the potential to make a clear contribution to the existing literature. The research design looks sound, and mainly builds on two well-established measures of readability. The analyses appear to be executed in a rigorous and transparent manner, and the authors generate a set of straightforward and intuitive but highly relevant findings. The reviewers hope that this manuscript will raise awareness, and encourage scientists and publishers to take readability into account when authoring or reviewing scientific papers.

Nevertheless, there are some issues that need to be addressed, and several revisions that would strengthen the manuscript. Most can be handled by editing the text to provide more details or clarity, or by improving the presentation of the figures. One will require a more fine-grained analysis of the data that is already available.

Essential Revisions:

1. Both of the readability formulas reported in the Materials and methods section contain mistakes. The constant in the Flesch Reading Ease (FRE) formula is incorrectly given as "206.385" instead of "206.835". The New Dale-Chall (NDC) Readability Formula is repeatedly missing the "× 100" that converts the fraction of "difficult words" to a percentage. The inequality symbols (i.e. > and {less than or equal to}) have also been transposed in the NDC formula meaning that, if followed as written, the scores would not be adjusted properly. From the data provided, it is clear that the correct NDC formula was used for the analyses. But we cannot check the FRE calculations because the "syllable counts" are missing from the data provided (see point 4). Please confirm that the correct FRE formula was used and fix the mistakes described above.

2. The readability measures chosen appear plausible and test two complementary dimensions of readability. However, please provide some more justification as to why these two measures were chosen and not others.

3. Please elaborate on the limitations of readability metrics in general, and those linked to text pre-processing. For example, visual aids are not captured even though they might improve comprehension; or the fact that FRE does not take into account parentheses and other ways of structuring text (which might also aid readability).

4. The authors have agreed to make the data and scripts openly accessible, which is applaudable. However, the data provided with the full submission is missing the syllable counts. Please provide all the data needed to recreate the figures in this version of the manuscript. This also includes the analyses related to the full texts, number of authors, and "general scientific jargon". Please see the section on "Source Data Files" in eLife's author guide for more details.

5. The current Figure 1 and Figure 3are packed with data and somewhat difficult to follow. Please divide them each into two separate figures. For Figure 1, separate the summary graphics from the panels showing the main trends. For Figure 3, the data related to number of authors should be separated from that related to "general scientific jargon".

6. The phrase "high-impact journals" is used without explanation. The specific meaning of the word "impact" in this context is unclear and could arguably be considered jargon. Please explain this phrase in the text or replace with a more accurate description like "highly cited journals".

7. Please elaborate on the criteria used to select the "six representative" open-access journals for the full-text analysis. Please also provide information equivalent to that provided for the journals in the abstract-only analysis.

8. As it stands now, the analysis remains at a very high level. Digging deeper into the existing data might reveal avenues to pursue to address the poor readability of scientific texts. Please report and discuss the trends in readability (or the average values) for the different journals and/or fields of research. For example, showing that articles in some journals are more readable than others might encourage someone to trace these differences back to specific editorial policies (word limits etc.) that could then be adopted more widely.

9. The literature on science communication is extensive, covering both the underlying communication process as well as its implications. Please strengthen the manuscript's links to this literature; for example, by discussing to what extent is it necessary that non-specialists can read and understand scientific articles, given that mainstream media and other new outlets have been created to specifically focus on non-specialist audiences in this context. While the reviewers likely agree with the authors' line of argument, some more contextualization and justification would be helpful here.

10. Please integrate the "Supplementary methods" into the "Materials and methods" section, to avoid the need to jump back and forth in the text. The sections on selecting journal fields and identifying "candidate jargon" would also benefit from being critically edited to improve clarity.

11. On a related point, please define classifications like "science-specific common words" and "candidate general science jargon" more clearly, and then consistently use them throughout the text. As currently written, some readers would not recognize uncommon/formal words - such as "novel", "robust", "moreover" and "therefore" - as scientific jargon. It would also be helpful if the discussion on this topic were expanded.

12. The FRE score of 48.2 for the submitted manuscript is respectable. However, please look for further opportunities to make the article easier to read, including ways that would not necessarily get picked up by the readability formulas referenced in the article. For example, while most English speakers are likely to know the verb "understand", fewer will be familiar with the adjective "understandable", and fewer still the noun "understandability" (which is used in the manuscript's abstract). Please consider using fewer nouns and more verbs throughout the abstract, main text and figure captions (e.g. "Scientific abstracts have become harder to read over time" rather than "Readability of scientific abstracts decreases over time".)

---

## [Author Response]

Essential Revisions:

*1. Both of the readability formulas reported in the Materials and methods section contain mistakes. The constant in the Flesch Reading Ease (FRE) formula is incorrectly given as "206.385" instead of "206.835". The New Dale-Chall (NDC) Readability Formula is repeatedly missing the "× 100" that converts the fraction of "difficult words" to a percentage. The inequality symbols (i.e. > and {less than or equal to}) have also been transposed in the NDC formula meaning that, if followed as written, the scores would not be adjusted properly. From the data provided, it is clear that the correct NDC formula was used for the analyses. But we cannot check the FRE calculations because the "syllable counts" are missing from the data provided (see point 4). Please confirm that the correct FRE formula was used and fix the mistakes described above.*

We thank the reviewers for noting the typos in the readability formulas.

These mistakes were not present in the analysis code and have now been corrected in the text. We further apologise for the syllable count being omitted from the source data we included. This has been fixed and included as source data (see the response to point 4 for a further details regarding the source data).

*2. The readability measures chosen appear plausible and test two complementary dimensions of readability. However, please provide some more justification as to why these two measures were chosen and not others.*

Our reason for using two measures was to make sure the result was not merely induced by the measure itself. The reasons for using FRE were because of its popularity and because it has been used in looking at temporal trends in other studies. The reason for using NDC was because it includes something different to FRE in how it is calculated (the common word lists). Further it performs comparatively compared to recently proposed readability measures. We have added some sentences to justify the choice of two readability metrics in the first paragraph of the “Language and Readability metrics” subsection of the Material and methods section.

*3. Please elaborate on the limitations of readability metrics in general, and those linked to text pre-processing. For example, visual aids are not captured even though they might improve comprehension; or the fact that FRE does not take into account parentheses and other ways of structuring text (which might also aid readability).*

We agree that it is important to emphasize the limitations of readability formulas. We have therefore amended the limitations paragraph with examples for text elements which contribute to understanding but which are often removed during pre-processing or not captured by readability formulas.

*4. The authors have agreed to make the data and scripts openly accessible, which is applaudable. However, the data provided with the full submission is missing the syllable counts. Please provide all the data needed to recreate the figures in this version of the manuscript. This also includes the analyses related to the full texts, number of authors, and "general scientific jargon". Please see the section on "Source Data Files" in eLife's author guide for more details.*

Source data is now provided for Figure 2, Figure 3, Figure 4 and Figure 6.

We have now added the summed syllable-count to the “Figure 2—source data 1”, which provides all the information for FRE and NDC to be calculated per article.

We have further added “number of authors” to the same file. We have added a similar file: “Figure 2—source data 2” for the analysis on the unpreprocessed data. This allows for full recreation of the relevant figures (Figure 2 and its supplement).

“Figure 3—source data 1” and “Figure 3—source data 2” is now included and will allow reproduction of Figure 3 and its supplement.

“Figure 4—source data 1” is now included and allows reproduction of Figure 4 and its supplements.

“Figure 6—source data 1” contains all the information needed to reproduce Figure 6.

“Figure 6—source data 2” is also included which allows for the analysis to be reproduced regarding the derivation of the science easy words.

Figure 5 and Figure 1 both use “Figure 2—source data 1” as their source data.

It is associated with Figure 2 as most of the data in the file belongs to Figure 2. In the figure captions of both Figure 1 and Figure 5 it is noted that “Figure 2—source data 1” is where the source data can be found.

In sum all figures with data now have relevant source data that can be used to recreate the figures. Finally, we have placed all the scripts on a public repository on GitHub: https://github.com/wiheto/readabilityinscience and these are now available.

*5. The current Figure 1 and Figure 3 are packed with data and somewhat difficult to follow. Please divide them each into two separate figures. For Figure 1, separate the summary graphics from the panels showing the main trends. For Figure 3, the data related to number of authors should be separated from that related to "general scientific jargon".*

We have divided up the figures as requested. There are now six figures in total: the summary graphic (Figure 1); the main readability result (Figure 2); the random effects per field (Figure 3; and per journal in the figure supplement); the full text data (Figure 4); the number of authors (Figure 5); the “jargon” analysis (Figure 6).

*6. The phrase "high-impact journals" is used without explanation. The specific meaning of the word "impact" in this context is unclear and could arguably be considered jargon. Please explain this phrase in the text or replace with a more accurate description like "highly cited journals".*

The reviewers are correct that this term is unclear, and this term is certainly a form of science-ese. The text has been amended to say “highly cited” both in the introduction and methods.

*7. Please elaborate on the criteria used to select the "six representative" open-access journals for the full-text analysis. Please also provide information equivalent to that provided for the journals in the abstract-only analysis.*

We have expanded upon the reasons for why these particular six journals have been included in the abstract-full text analysis. Importantly, one of the main reasons to choose these journals was because they were Open Access - and Open Access journals are the only ones that allow us to download full texts en masse without violating the terms and conditions of services like Pubmed. We have added a supplementary file to the manuscript (Supplementary file 4), which displays information about the journals included in this analysis. Further, we have also now provided additional supplementary figures (Figure 4—figure supplement 2), demonstrating that the observed correlation between readability for abstracts and full texts is present in each of the journals included in the analysis.

*8. As it stands now, the analysis remains at a very high level. Digging deeper into the existing data might reveal avenues to pursue to address the poor readability of scientific texts. Please report and discuss the trends in readability (or the average values) for the different journals and/or fields of research. For example, showing that articles in some journals are more readable than others might encourage someone to trace these differences back to specific editorial policies (word limits etc.) that could then be adopted more widely.*

We agree with the reviewers that a deeper analysis can yield ideas and hypotheses for future research on how to improve readability. We have therefore extracted the random effect (i.e. slope) of each journal from the mixed models and plotted them both by themselves, and as field-summaries, in Figure 3 and its supplement. This shows that the trend is present in all fields, although some journals do not show a decline in readability. We have amended the results section accordingly, and added a discussion about the possibility of future research to examine why there are differences between fields and journals, and how this could yield clues on how to address the problem.

*9. The literature on science communication is extensive, covering both the underlying communication process as well as its implications. Please strengthen the manuscript's links to this literature; for example, by discussing to what extent is it necessary that non-specialists can read and understand scientific articles, given that mainstream media and other new outlets have been created to specifically focus on non-specialist audiences in this context. While the reviewers likely agree with the authors' line of argument, some more contextualization and justification would be helpful here.*

In answer to this point we have rewritten two paragraphs. The paragraph that begins “Lower readability implies less accessibility, particularly for non-specialists, such as journalists, policy-makers and the wider public” (Discussion, fourth paragraph) now clarifies that “non-specialists” also applies to journalists within the mainstream media. This should clarify that this even affects those that use science journalism as a news source. We have also changed the focus of this paragraph from previously talking about scientific literacy and now focus more on scientific journalism. This is because it is hard to see that scientific literacy is causally affected by our results here, and it was a speculative link of troubling trends.

Additionally the paragraph that begins “What more can be done to reverse this trend?” (Discussion, penultimate paragraph) makes greater links with the literature in scientific communication. We state some of the more prominent proposals that have been made to make scientific texts more accessible.

*10. Please integrate the "Supplementary methods" into the "Materials and methods" section, to avoid the need to jump back and forth in the text. The sections on selecting journal fields and identifying "candidate jargon" would also benefit from being critically edited to improve clarity.*

The two sections that were previously in the Supplementary Methods, namely the selection of journal fields and identifying candidate jargon have both been moved to the Materials and methods section. Both sections have been made shorter and clearer.

*11. On a related point, please define classifications like "science-specific common words" and "candidate general science jargon" more clearly, and then consistently use them throughout the text. As currently written, some readers would not recognize uncommon/formal words - such as "novel", "robust", "moreover" and "therefore" - as scientific jargon. It would also be helpful if the discussion on this topic were expanded.*

The Material and methods subsection “Science-specific common word and general science jargon lists” has been completely rewritten where we give clear definitions of what these two terms mean. We have also dropped the term “candidate” from the entire text as it could be considered confusing. The initial reason for including it was to not definitively declare these terms as jargon, since they have been defined only by excluding certain categories of words. An additional sentence has been added to the discussion paragraph to clarify how to interpret general scientific jargon.

*12. The FRE score of 48.2 for the submitted manuscript is respectable. However, please look for further opportunities to make the article easier to read, including ways that would not necessarily get picked up by the readability formulas referenced in the article. For example, while most English speakers are likely to know the verb "understand", fewer will be familiar with the adjective "understandable", and fewer still the noun "understandability" (which is used in the manuscript's abstract). Please consider using fewer nouns and more verbs throughout the abstract, main text and figure captions (e.g. "Scientific abstracts have become harder to read over time" rather than "Readability of scientific abstracts decreases over time".)*

We have now gone through the manuscript and, to the best of our ability, tried to improve the readability further. For example, sentences have been shortened, nouns have been replaced by verbs when suitable, and complicated words have been replaced by simpler synonyms when this could be done without losing meaning. Further, some parts of the text have been completely rewritten to make them easier to understand, such as the entire journal selection procedure paragraph in the Material and methods section. Please see the track-changed document for all readability improvements in the manuscript.

In addition to the revisions detailed above requested by the reviewers, we identified three additional issues with the article:

There was a typo in the FRE calculation of a scalar in the analysis code. The part “1.015 * words/sentences” was incorrectly set to 1.1015. This has led to a very slight increase in all FRE scores when fixed. This change has had no effect on the observed trend. However, the sentence “In other words, more than a quarter of scientific abstracts now have a readability considered beyond college graduate level English.” (Discussion, first paragraph) has been changed to “In other words, more than a fifth of scientific abstracts now have a readability considered beyond college graduate level English.”

The selection criteria state that 12 journals from each field were included. However, for the field Immunology, only 11 journals had been included by mistake. The Journal “Brain Behav Immun” has now been added. This has increased the number of abstracts from 707,452 to 709,577 and all values reported in the article have been updated accordingly. This includes a new training and verification set for the “general science jargon” analysis, for which there are now 2,138 general jargon words instead of 2,140 previous ones. These changes have not led to any difference in the interpretation of the results.

We noticed that the language parameters for the TreeTagger software used for preprocessing differed slightly between Linux and Windows versions. Previously the full text analysis had been run on Windows and the abstract analysis on Linux. The full text analysis has now also been run on Linux. This has led to minor differences in readability scores for the full text analysis. These minor differences have no impact on the interpretation of the results. We have also noted relevant package version numbers and the operating system used in the Material and methods section.

In sum these three changes have no impact on the conclusions drawn from the overall analysis. They however ensure that the analysis is more correct and easier to reproduce.